# Remote versus face-to-face neuropsychological testing for dementia research: a comparative study in people with Alzheimer's disease, frontotemporal dementia and healthy older individuals

Maï-Carmen Requena-Komuro ![ORCID] [1,2] Jessica Jiang,[1] Lucianne Dobson,[1] Elia Benhamou,[1,3] Lucy Russell,[1] Rebecca L Bond,[1] Emilie V Brotherhood,[1] Caroline Greaves,[1] Suzie Barker,[1] Jonathan D Rohrer ![ORCID] ,[1] Sebastian J Crutch,[1] Jason D Warren ![ORCID] ,[1] Chris JD Hardy ![ORCID] [1]

M-CR-K and JJ contributed equally.

M-CR-K and JJ are joint first authors.

For numbered affiliations see end of article.

**Correspondence to**
Dr Chris JD Hardy;
chris.hardy@ucl.ac.uk

## ABSTRACT

**Objectives** We explored whether adapting neuropsychological tests for online administration during the COVID-19 pandemic was feasible for dementia research.

**Design** We used a longitudinal design for healthy controls, who completed face-to-face assessments 3–4 years before remote assessments. For patients, we used a cross-sectional design, contrasting a prospective remote cohort with a retrospective face-to-face cohort matched for age/education/severity.

**Setting** Remote assessments were conducted using video-conferencing/online testing platforms, with participants using a personal computer/tablet at home. Face-to-face assessments were conducted in testing rooms at our research centre.

**Participants** The remote cohort comprised 25 patients (n=8 Alzheimer's disease (AD); n=3 behavioural variant frontotemporal dementia (bvFTD); n=4 semantic dementia (SD); n=5 progressive non-fluent aphasia (PNFA); n=5 logopenic aphasia (LPA)). The face-to-face patient cohort comprised 64 patients (n=25 AD; n=12 bvFTD; n=9 SD; n=12 PNFA; n=6 LPA). Ten controls who previously participated in face-to-face research also took part remotely.

**Outcome measures** The outcome measures comprised the strength of evidence under a Bayesian framework for differences in performances between testing environments on general neuropsychological and neurolinguistic measures.

**Results** There was substantial evidence suggesting no difference across environments in both the healthy control and combined patient cohorts (including measures of working memory, single-word comprehension, arithmetic and naming; Bayes Factors (BF)$_{01}$ >3), in the healthy control group alone (including measures of letter/category fluency, semantic knowledge and bisyllabic word repetition; all BF$_{01}$ >3), and in the combined patient cohort alone (including measures of working memory, episodic memory, short-term verbal memory, visual perception, non-word reading, sentence comprehension and bisyllabic/trisyllabic word repetition; all BF$_{01}$ >3). In the control cohort alone, there was substantial evidence in support of a difference across environments for tests of visual perception (BF$_{01}$=0.0404) and monosyllabic word repetition (BF$_{01}$=0.0487).

**Conclusions** Our findings suggest that remote delivery of neuropsychological tests for dementia research is feasible.

## STRENGTHS AND LIMITATIONS OF THIS STUDY

⇒ Diverse patient cohorts representing rare dementias with specific communication difficulties.
⇒ Sampling of diverse and relevant neuropsychological domains.
⇒ Use of Bayesian statistics to quantify the strength of evidence for the putative null hypothesis (no effect between remote and face-to-face testing).
⇒ Relatively small cohort sizes.
⇒ Lack of direct head-to-head comparisons of test environment in the same patients.

## INTRODUCTION

The COVID-19 pandemic and associated social distancing and lockdown measures imposed a series of daunting challenges for conducting research with people with dementia. In the UK, three national lockdowns between March 2020 and February 2021 largely prevented face-to-face research. People with dementia are at increased risk of COVID-19,[1] and many participants understandably did not feel safe to travel for research, particularly before widespread vaccination was implemented. Here, we

describe our attempts to translate our traditional neuropsychological and neurolinguistic test batteries (typically administered face-to-face) for remote administration.

Development and implementation of online cognitive assessments for patients with dementia, particularly within communities who experience difficulties in accessing clinical care are not new.[2] Telemedicine has been previously used successfully in Alzheimer's disease (AD)[3 4] and with rarer dementias such as primary progressive aphasia (PPA)[5–7] and behavioural variant frontotemporal dementia (bvFTD).[7] However, due to COVID-19, there has been a more pervasive shift towards the use of online methods to meet clinical, support and research needs.[8 9]

A review by Hunter and colleagues[10] summarises 20 years of research comparing face-to-face and online administration of cognitive tests in healthy older adults (≥40 years old) and participants diagnosed with mild cognitive impairment, AD or other types of dementia (often unspecified). The authors identified 12 studies that used video-conferencing methods. Overall, there was clear evidence to suggest that remote cognitive testing for people living with AD and other forms of dementia is feasible. Additionally, there is evidence to suggest that online performance remains stable over time (with a maximum delay of 3 months between assessments), particularly for the domains of executive function, working memory, verbal episodic memory and language. Minimal evidence was available for visuospatial tasks, and tests of single word and sentence comprehension.

Notwithstanding considerable progress in this area to date, further research into the feasibility of remote neuropsychological testing of patients with neurodegenerative diseases is required. There are three main considerations that need addressing: (1) Adaptation: are similar performance outcomes obtained on neuropsychological tests designed for face-to-face administration when given remotely? If performance across modalities is equivalent, then this could allow for pooling of data collected face-to-face and remotely, potentially allowing more equitable access to research for participants who are not able to physically travel to research centres. (2) Demand: to what extent are research participants with and without dementia willing to engage in remote neuropsychological research? (3) Acceptability: how satisfactory is remote testing for research participants including those with diverse forms of dementia?[11]

Based largely on the face-to-face protocol for general neuropsychological and neurolinguistic testing used at our research centre, we built a protocol for remote testing of patients diagnosed with typical AD, patients representing major variants of PPA (semantic dementia (SD), progressive non-fluent aphasia (PNFA), logopenic aphasia (LPA)) and bvFTD. Patients were tested from their homes via the widely used video-conferencing software, Zoom (Zoom Video Communications). We also recruited a small cohort of healthy older adults who had taken part in our face-to-face research at the Dementia Research Centre 3–4 years before the pandemic. Here, we compared the healthy controls' performance on several neuropsychological and neurolinguistic tests between the two testing environments (face-to-face vs remote). We also compared the performance of patients tested remotely with a historical face-to-face cohort of patients chosen to represent the same syndromes and to match the remote cohort based on age, education and symptom duration. We adopted a Bayesian approach that assesses the amount of evidence in favour of the null hypothesis (ie, that there is no significant difference in performance on a given neuropsychological task between testing environments) relative to the alternative hypothesis (ie, that there is a significant difference in performance on a given neuropsychological task between testing environments).

Following previous research,[10] we did not predict major differences in terms of participants' performances when tested face-to-face and remotely on most neuropsychological and neurolinguistic tests. However, we did consider the potential for poorer performance on tests of speech perception that were administered remotely, given additional difficulties associated with controlling the remote auditory environment.

## METHODS
### Participant recruitment and group matching
Recruitment for the study took place between February and August 2021. Potential patient participants were identified via the Specialist Cognitive Disorders Clinic at the National Hospital for Neurology and Neurosurgery, direct research referrals from external clinicians or via Rare Dementia Support (www.raredementiasupport.org); healthy controls were recruited via our research participant database. Eighty-seven potentially eligible participants were identified, and all of these were contacted.

An initial telephone screen was conducted for each participant to establish they had access to the necessary equipment (tablet or desktop/laptop computer), a broadband internet connection, a quiet testing space to support the remote research assessment, and no preclusive hearing or visual impairments. We also performed the telephone version of the Mini-Mental State Examination (T-MMSE) with patients to assess their disease severity.[12 13] A minimum score of 12 on the T-MMSE (which corresponds to a converted MMSE score of 16) was used as an inclusion criterion.[12] No participants were excluded after the telephone screen.

Twenty-five patients (eight with typical AD, three bvFTD, four SD, five PNFA, five LPA) were recruited for the remote study. For comparison purposes, a reference historical cohort comprising 64 patients (25 with AD, 12 bvFTD, 9 SD, 12 PNFA, 6 LPA) who had undertaken a face-to-face research assessment at our centre between 2013 and 2020 was selected, matching the cohort assessed remotely as closely as possible for syndromic composition, age, years of education and symptom duration. Henceforth, these are referred to as the 'remote' and

'face-to-face' patient cohorts, respectively. All patients fulfilled consensus diagnostic criteria for the relevant syndromic diagnosis[14–16] and all had clinically mild-to-moderate severity disease. Where available, brain MRI was consistent with the syndromic diagnosis, without evidence of significant cerebrovascular burden. Ten healthy older individuals with no history of neurological or psychiatric illness and who had been seen for face-to-face testing 3–4 years previously also underwent remote assessments. The neuropsychological tests reported in this research article were not used for diagnostic purposes. Demographic and clinical details for all participants are summarised in table 1. All participants gave informed consent for their involvement in the study.

### Testing procedure: face-to-face

Data for the reference historical cohort were collected under our face-to-face research assessment protocol, as delivered in experimental sessions at the Dementia Research Centre between 2013 and 2020. Under this protocol, all neuropsychological tests were administered in dedicated quiet testing rooms, with the participant sitting opposite the experimenter. Patients were predominantly tested on their own, unless the informant accompanying them to the study visit requested to be present and the participant agreed to this. In these cases, the informant was explicitly asked not to intervene during testing. No feedback was given on performance and no time limits were imposed (unless timing was intrinsic to the test). A battery of general neuropsychological and neurolinguistic tests (see tables 2 and 3) were administered, following standard methods. The neurolinguistic test was developed specifically to characterise the language profiles of people with PPA and therefore, was not administered to participants with bvFTD or AD.

### Modifying the face-to-face battery of neuropsychological tests for remote delivery

We reviewed the battery of general neuropsychological and neurolinguistic tests that had been used historically at our centre for face-to-face administration, in order to identify tests that could be feasibly delivered remotely online while preserving the overall structure of the tests and sampling across cognitive domains as far as possible (see online supplemental table 1). Where a task required visual stimulus presentation, a high-quality copy of the stimuli was made. Images were then imported into Microsoft Power-Point for subsequent presentation to the participant via screen share.

Tests that were retained for remote testing (ie, tests administered to both the remote and face-to-face patient cohorts) are itemised in tables 2 and 3. Where applicable, we sought permission from the test publishers to adapt tests for remote administration.

### Testing procedure: remote

An initial session was conducted via Zoom to accustom participants to the remote testing format, check the screen and sound sharing options on Zoom, and that the quality of their internet connection was acceptable. Technical aspects of the set-up for remote testing are detailed in online supplemental text and online supplemental figure 1.

The remote neuropsychological and neurolinguistic tests each took around an hour to administer. To minimise fatigue,[17] tests were delivered in separate testing sessions typically within a week (and never more than 2 weeks apart).

### Feedback on remote testing experience

At the end of each remote testing session, the experimenter debriefed each participant. This provided them with the opportunity to raise any technical issues, give their impressions of the remote testing session and note any distractions that may have occurred for them.

Where time allowed, at the end of the session, participants were also asked by the experimenter to indicate on a 10-point integer scale how comfortable they had felt with the remote testing format, with 10 indicating 'very comfortable'.

### Statistical analysis

All statistical analyses were performed in JASP (V.0.16).

The remote and face-to-face patient cohorts were compared on demographic characteristics using independent samples t-tests and Wilcoxon rank-sum tests. Healthy controls' scores in remote and face-to-face testing environments were compared using paired samples t-tests or (where the assumption of normality was not met) Wilcoxon signed-rank tests. Healthy controls' and patients' ratings of comfort after the general neuropsychological and neurolinguistic sessions with the remote testing set-up were compared using Wilcoxon rank-sum tests. To reduce type I error, no corrections for multiple comparisons were applied.

We did not perform between-group comparisons of neuropsychological and neurolinguistic performance as these syndromic profiles of the neuropsychological and neurolinguistic tests have been reviewed and published previously.[18 19]

In comparing testing environments, our null hypothesis was that there would be no effect of testing environment on neuropsychological performance—that is, no differences in performance between remote and face-to-face assessment settings—for any participant group. To critically assess the magnitude of evidence in favour of this null hypothesis versus the alternative hypothesis (ie, that there was in fact an effect of testing environment) particularly in light of the relatively small patient cohorts here, we employed a Bayesian approach.[20] Bayesian independent samples t-tests (and non-parametric equivalents where assumptions of the general linear model were violated) were performed for each general neuropsychological

**Table 1** General demographic, clinical and environmental characteristics for all participant groups: comparison of remote and face-to-face cohort characteristics

| Characteristic | CTL Remote | AD F2F | AD Remote | bvFTD F2F | bvFTD Remote | SD F2F | SD Remote | PNFA F2F | PNFA Remote | LPA F2F | LPA Remote |
|---|---|---|---|---|---|---|---|---|---|---|---|
| N | 10 | 25 | 8 | 12 | 3 | 9 | 4 | 12 | 5 | 6 | 5 |
| Gender (M/F) | 7/3 | 15/10 | 5/3 | 8/4 | 3/0 | 7/2 | 2/2 | 6/6 | 2/3 | 4/2 | 4/1 |
| Age (years) | 74.0 (4.1)* | 69.5 (4.1) | 69.8 (5.9) | 70.3 (2.9) | 72.0 (5.6) | 60.3 (5.2) | 58.3 (9.3) | 67.1 (5.2) | 68.2 (6.4) | 69.5 (3.6) | 71.6 (4.7) |
| Education (years) | 17.6 (0.7) | 14.8 (2.5) | 16.0 (3.7) | 14.9 (2.7) | 14.7 (3.1) | 16.0 (1.9) | 15.5 (3.1) | 15.3 (2.3) | 14.8 (2.8) | 16.0 (1.7) | 16.0 (3.3) |
| Handedness (R/L/A) | 9/1/0 | 23/1/1 | 8/0/0 | 11/1/0 | 3/0/0 | 9/0/0 | 3/1/0 | 10/2/0 | 5/0/0 | 5/1/0 | 5/0/0 |
| Symptom duration (years) | NA | 6.8 (2.3) | 7.3 (3.7) | 4.7 (2.1) | 3.7 (3.8) | 4.1 (1.9) | 3.5 (1.9) | 3.4 (1.7) | 3.2 (0.8) | 4.2 (1.9) | 3.8 (1.9) |
| Desktop/laptop/tablet | 4/5/1 | NA | 3/2/3 | NA | 1/2/0 | NA | 1/3/0 | NA | 1/2/2 | NA | 1/3/1 |
| Study partner present (Y/N) | NA | NA | 5/3 | NA | 2/1 | NA | 2/2 | NA | 2/3 | NA | 3/2 |
| Interruptions (N) | 1 | NA | 1 | NA | 1 | NA | 0 | NA | 0 | NA | 0 |

Mean (SD) values are shown for each group. No significant differences between remote and face-to-face cohorts were found.

*On average, healthy controls were 3.5 years younger when tested face-to-face.

A, ambidextrous AD, patient group with typical Alzheimer's disease; bvFTD, patient group with behavioural variant frontotemporal dementia; CTL, healthy control group; F, female; F2F, face-to-face; L, left; LPA, patient group with logopenic progressive aphasia; M, male; N, number of participants per group; NA, not applicable; PNFA, patient group with progressive non-fluent aphasia; R, right; SD, patient group with semantic dementia.

**Table 2** General neuropsychological performance for all participant groups: comparison of remote and face-to-face test administration

| | CTL | | All patients | | AD | | bvFTD | | SD | | PNFA | | LPA | |
|---|---|---|---|---|---|---|---|---|---|---|---|---|---|---|
| | F2F | Remote | F2F | Remote | F2F | Remote | F2F | Remote | F2F | Remote | F2F | Remote | F2F | Remote |
| **Number tested** | 10 | 10 | 64 | 25 | 25 | 8 | 12 | 3 | 9 | 4 | 12 | 5 | 6 | 5 |
| General intellect | | | | | | | | | | | | | | |
| WASI matrix (/32) | 26.9 (2.4) | 26.3 (3.5) | **16.5 (8.86)** | *17.7 (9.24)* | **14.5 (7.5)** | *13.7 (11.5)*[b] | 14.6 (9.5)[a] | 7.7 (2.5) | 25.8 (4.9) | 23.3 (3.4) | 18.6 (8.3)[a] | 24.0 (2.6) | 10.0 (9.2) | 17.6 (10.3) |
| Episodic memory | | | | | | | | | | | | | | |
| RMT faces (short) (/25) | NA | 23.3 (2.5) | 18.9 (3.68) | 19.2 (4.21) | 18.9 (3.7)[b] | 15.3 (2.1) | NA | 20.3 (4.0) | NA | 19.3 (3.7) | NA | 22.4 (3.7) | NA | 21.6 (3.8) |
| RMT faces (/50) | 42.7 (4.7) | NA | 32.6 (7.40) | NA | 30.44 (6.56)[g] | NA | 28.56 (6.29)[c] | NA | 31.5 (4.47)[a] | NA | 40.3 (5.79)[b] | NA | 33.17 (8.93) | NA |
| Working memory | | | | | | | | | | | | | | |
| DS (reverse) (/12) | **8.0 (2.6)** | **8.2 (2.2)** | **4.82 (2.47)** | 4.64 (2.40) | 4.6 (1.9)[c] | 4.0 (1.9) | 5.0 (3.1)[a] | 4.0 (0.0) | 6.9 (2.7) | 5.5 (2.7) | 4.4 (1.7)[d] | 5.2 (4.2) | 2.4 (1.1)[a] | 4.8 (1.8) |
| Short-term verbal memory | | | | | | | | | | | | | | |
| DS (forward) (/12) | 9.8 (2.0) | 9.2 (2.0) | 6.68 (2.94) | 6.24 (2.52) | **6.5 (2.1)**[a] | 6.6 (2.3) | 7.8 (2.9) | 6.3 (0.6) | 9.4 (2.2) | 7.3 (0.5) | 5.1 (2.6)[b] | 6.0 (4.4) | 3.0 (3.1)[a] | 5.0 (2.5) |
| Language | | | | | | | | | | | | | | |
| BPVS (/150) | **149.0 (1.1)** | **148.9 (1.1)** | **120 (39.3)** | 134 (26.4) | **135.1 (24.0)**[a] | **133.9 (21.4)** | 118.7 (28.9)[a] | 148.0 (1.0) | 61.9 (44.1) | 98.0 (46.3) | 136.3 (30.4) | 142.8 (8.4) | 120.2 (39.6) | 145.2 (4.4) |
| GNT (/30) | 27.3 (2.1) | 26.8 (1.8) | **11 (9.33)** | 13 (8.62) | **13.7 (8.5)**[a] | **12.4 (6.3)** | 11.5 (11.0)[a] | 20.0 (4.4) | 0.2 (0.4) | 0.5 (1.0) | 15.8 (6.8) | 20.2 (7.6) | 6.2 (7.7) | 12.4 (7.0) |
| NART (/50) | 45.2 (3.4) | 44.6 (3.0) | 26.1 (13.4) | 31.4 (13.1) | 28.3 (11.9)[a] | 37.1 (4.6) | 30.7 (13.1) | 40.0 (2.0) | 16.1 (10.2) | 16.3 (13.6) | 28.3 (15.0)[f] | 30.8 (14.9) | 20.4 (18.5)[a] | 29.8 (16.9) |
| Category fluency (60 s, 'animals') | **25.0 (6.6)** | **24.7 (5.4)** | 8.46 (5.75) | 12.2 (8.10) | **10.0 (5.6)** | **10.4 (6.7)** | 8.0 (6.1) | 10.3 (6.0) | 6.1 (5.3)[a] | 9.8 (6.2) | 8.6 (5.8)[d] | 18.6 (11.5) | 5.7 (5.8) | 11.6 (8.1) |
| Arithmetic | | | | | | | | | | | | | | |
| GDA total (/24) | **16.6 (6.4)** | **16.1 (4.0)** | **6.90 (6.25)** | **6.30 (5.60)** | **4.9 (5.0)**[e] | **4.8 (5.4)** | 8.2 (7.4)[a] | 8.3 (7.2) | 13.3 (5.5) | 7.3 (4.0) | 5 (3.6)[d] | 6.4 (7.2) | 1.7 (1.5) | 7 (5.8) |
| Visuospatial | | | | | | | | | | | | | | |
| VOSP OD (/20) | 19.5 (0.7) | 18.4 (1.2) | **15.6 (3.77)** | 15.2 (3.52) | 16.1 (2.6)[a] | 13.0 (3.0) | 14.0 (4.2) | 17.0 (1.0) | 15.3 (6.3)[a] | 14.5 (5.9) | 16.8 (3.6)[b] | 17.6 (2.1) | 15.8 (3.1) | 15.8 (2.7) |
| Executive | | | | | | | | | | | | | | |
| Letter fluency (60 s, 'F') | **19.7 (6.0)** | **20.8 (4.2)** | 7.70 (5.27) | 11.3 (6.29) | 10.3 (5.2) | 11.9 (7.4) | 6.8 (3.9)[a] | 7.3 (2.3) | 7.0 (3.6)[b] | 10.8 (7.0) | 4.8 (5.8)[d] | 14.0 (8.2)[a] | 2.6 (3.8)[a] | 11.3 (3.9) |

Mean (SD) values of performance on neuropsychological tests are shown (maximum scores are indicated in parentheses) for each testing environment (face-to-face vs remote research setting) for each patient group. **Bold** indicates values for which there was substantial evidence in support of the null hypothesis (H0); *italics* indicate values for which there was substantial evidence in support of the alternative hypothesis (H1). Exact values for the Bayes factor comparing H0 against H1 (BF[01]) are presented in online supplemental table 2. The general neuropsychological tests comprised tests of general intellect (WASI) matrix reasoning,[39] episodic memory (RMT) for faces,[40] working memory (DS forwards)[41] language (GNT,[42] BPVS[43] and NART[44]), arithmetic (GDA),[44] visual perception (VOSP OD task[46]) and executive function (DS reverse,[41] letter ('F') and category fluency ('animal') tasks[47]). A reduced number of participants completed certain tests, as follows: [a]n–1, [b]n–2, [c]n–3, [d]n–4, [e]n–5, [f]n–6, [g]n–9, [h]n–16.
All patients=combined patient cohort.
AD, patient group with typical Alzheimer's disease; BPVS, British Picture Vocabulary Scale; bvFTD, patient group with behavioural variant frontotemporal dementia CTL, healthy control group; DS, digit span; F2F, face-to-face; GDA, Graded Difficulty Arithmetic test; GNT, Graded Naming Test; LPA, patient group with logopenic progressive aphasia; NA, not available; NART, National Adult Reading Test; OD, object decision; PNFA, patient group with progressive non-fluent aphasia; RMT, Recognition Memory Test; SD, patient group with semantic dementia; VOSP, Visual Object and Space Perception; WASI, Wechsler Adult Intelligence Scale.

**Table 3** Neurolinguistic performance for all participant groups: comparison of face-to-face and remote test administration

| Test | CTL | | All patients | | SD | | PNFA | | LPA | |
|---|---|---|---|---|---|---|---|---|---|---|
| | F2F | Remote | F2F | Remote | F2F | Remote | F2F | Remote | F2F | Remote |
| **Number tested** | 10 | 10 | 27 | 14 | 9 | 4 | 12 | 5 | 6 | 5 |
| Phoneme perception | | | | | | | | | | |
| PALPA-3 (/36) | 34.9 (1.6) | 33.8 (2.0) | 33.5 (4.28) | 33.1 (2.85) | 35.3 (1.3) | 33.5 (2.6) | 31.9 (5.7) | 34.0 (1.6) | 33.8 (2.4)[a] | 32.0 (4.0) |
| Reading | | | | | | | | | | |
| Non-word reading (/25) | 24.4 (0.8) | 24.9 (0.3) | **18.1 (5.51)** | **16.1 (8.58)** | 19.6 (4.9) | 16.0 (9.8) | 19.3 (5.7)[d] | 15.8 (8.7) | 14.5 (5.4) | 16.4 (9.6) |
| Regular reading (/25) | 25.0 (0.0) | 25.0 (0.0) | 23.3 (2.83) | 21.8 (5.07) | 22.8 (3.9)[d] | 21.3 (6.2) | 23.3 (2.2)[g] | 22.2 (4.2) | 24.5 (0.7)[d] | 21.8 (6.1) |
| Irregular reading (/25) | 24.7 (0.7) | 25.0 (0.0) | 19.6 (5.30) | 21 (5.39) | 18.8 (5.8)[d] | 18.3 (8.9) | 19.8 (6.4)[g] | 21.6 (4.4) | 21.5 (3.5)[d] | 22.6 (2.1) |
| Naming | | | | | | | | | | |
| BNT (/30) | **29.2 (0.8)** | **29.0 (1.0)** | 13.4 (10.9) | 17.4 (9.58) | 3.3 (3.4) | 9.5 (7.1) | 24.7 (4.4)[b] | 24.4 (7.6) | 9.7 (8.0) | 16.8 (8.8) |
| Semantic association | | | | | | | | | | |
| Camel and cactus (/32) | **31.0 (1.0)** | **30.9 (1.1)** | 25.4 (5.58) | 28.4 (2.71) | 22.3 (5.1)[e] | 25.5 (2.6) | 28.3 (5.5)[h] | 30.0 (1.4) | 29.0 (0.0)[e] | 29.0 (2.1) |
| Word comprehension | | | | | | | | | | |
| Concrete synonyms (/25) | 24.5 (0.5) | 24.3 (0.5) | **18.7 (5.99)** | **20.2 (4.00)** | 15.5 (2.9)[c] | 17.0 (4.0)[a] | 22.2 (3.2)[b] | 21.0 (4.2) | 15.6 (9.4)[a] | 21.4 (3.5) |
| Abstract synonyms (/25) | 24.8 (0.4) | 24.3 (0.8) | 18.5 (6.20) | 20.6 (3.12) | 14.4 (1.5)[d] | 17.3 (1.2)[a] | 21.4 (3.7)[b] | 21.2 (3.4) | 16.5 (11.2)[b] | 22.3 (2.1)[a] |
| Sentence comprehension | | | | | | | | | | |
| PALPA-55 (/24) | 23.8 (0.4) | 23.1 (1.3) | **20.5 (3.83)** | **20.9 (2.28)** | 22.6 (1.6)[a] | 21.8 (2.2) | 19.6 (4.8)[a] | 21.0 (2.6) | 18.8 (2.6)[a] | 20.0 (2.1) |
| Speech repetition | | | | | | | | | | |
| Monosyllabic word repetition (/15) | *14.6 (0.5)* | *12.7 (1.6)* | 13.8 (1.92) | 10.6 (3.37) | 14.8 (0.4) | 10.8 (3.2) | 12.8 (2.6)[c] | 10.0 (4.2) | 14.0 (1.2)[b] | 11.2 (3.3) |
| Bisyllabic word repetition (/15) | **14.7 (0.5)** | **14.7 (0.7)** | **13.5 (2.65)** | **13.4 (2.95)** | 14.8 (0.7) | 13.0 (1.6) | 12.8 (3.3)[c] | 12.4 (4.7) | 12.0 (2.9)[b] | 14.8 (0.4) |
| Trisyllabic word repetition (/15) | 15.0 (0.0) | 15.0 (0.0) | **12.8 (4.18)** | **12.9 (3.89)** | 14.8 (0.4) | 12.8 (3.2) | 11.1 (5.9)[c] | 12.0 (6.2) | 12.3 (2.9)[b] | 14.0 (1.0) |
| Graded difficulty sentence repetition (/10) | 9.8 (0.4) | 9.3 (0.9) | 6.37 (2.89) | 5.79 (2.36) | 8.3 (1.2) | 7.3 (1.7) | 4.6 (3.0)[e] | 5.2 (3.2) | 4.7 (2.9)[c] | 5.2 (1.6) |
| Sentence construction | | | | | | | | | | |
| Spoken sentences (/25) | 25.0 (0.0) | 24.9 (0.3) | 18.6 (6.70) | 22.6 (3.34) | 19.1 (7.5) | 23.5 (1.3) | 18.0 (7.4)[d] | 21.6 (5.1) | 18.0 (5.3)[c] | 22.8 (2.7) |

Mean (SD) values of performance on neurolinguistic tests are shown (maximum scores are indicated in parentheses) for each participant group. **Bold** indicates values for which there was substantial evidence in support of the null hypothesis (H0); *italics* indicate values for which there was substantial evidence in support of the alternative hypothesis (H1). Exact values for the Bayes factor comparing H0 against H1 ($BF_{01}$) are presented in online supplemental table 3. The neurolinguistic tests comprised tests of phoneme perception (a shortened version of the PALPA Test 3–minimal pair discrimination[48]), reading (graded non-word reading,[49] and graded tests of regular word and irregular word reading adapted from Coltheart *et al*[50]), confrontation naming (a subset of items from the BNT[51]), semantic association (modified camel and cactus test[52]), single-word comprehension (concrete and abstract synonyms[53], sentence comprehension (a shortened version of the PALPA subtest 55[48]), speech repetition (tests of monosyllabic, bisyllabic and trisyllabic single-word repetition[54], graded difficulty sentence repetition using a subset of items from the sentence repetition test[55] and sentence construction based on Rohrer *et al*.[56] We opted to include two naming tasks as the GNT is part of the core neuropsychological tests at our centre and therefore administered to patients with all diagnoses. However, as patients with SD often score at floor on this task,[57] we also include a shortened version of the BNT in our neurolinguistic test that is administered to patients with PPA as patients with SD are more likely to be above floor on this task.[26] A reduced number of participants completed certain tests: [a]n–1, [b]n–2, [c]n–3, [d]n–4, [e]n–5, [f]n–6, [g]n–8, [h]n–9. In the spoken sentence construction test, participants are given five different words (walked, radio, throw, green, tree) and are asked to produce a sentence that contains each word in turn. Sentences are then scored based on grammar and semantic sense, each sentence counting for a maximum of 5 points.
All patients=combined patient cohort.
BNT, Boston Naming Test; CTL, healthy control group F2F, face-to-face; GNT, Graded Naming Test; LPA, patient group with logopenic progressive aphasia; N, number of participants per group; PALPA, Psycholinguistic Assessments of Language Processing in Aphasia; PNFA, patient group with progressive non-fluent aphasia; PPA, primary progressive aphasia; SD, patient group with semantic dementia.

and neurolinguistic test in each patient group separately. As numbers in some groups were quite small, we also conducted analyses for a combined patient cohort in both environments. Healthy control performance was compared using Bayesian paired samples t-tests (or appropriate non-parametric equivalent). A Bayes factor, which is the ratio of evidence supporting the null hypothesis over the alternative hypothesis (hereafter $BF_{01}$), was calculated for each comparison using JASP. A $BF_{01}$ value >3 indicates substantial evidence in favour of the null hypothesis while a value <0.33 supports the alternative hypothesis; $BF_{01}$ values between 0.33 and 3 are classified as 'anecdotal' evidence, comparable with non-significant differences in inferential statistics.[21 22] Bayes factor values are presented in online supplemental tables 2 and 3.

In comparing groups on comfort ratings after the remote sessions, our null hypothesis was that there would be no differences in comfort ratings between healthy controls and patients; our alternative hypothesis was that healthy control participants would report higher comfort ratings than patients.

Finally, we conducted F-tests and Levene's equality of variance tests to evaluate differences in variability between the two testing environments.

### Patient and public involvement

In August and September 2020, we contacted 527 people (comprising healthy control participants and people with a diagnosis of a dementia) who had previously taken part in our face-to-face research programmes in the Dementia Research Centre, University College London, or who had expressed an interest in doing so in the future. They were asked, 'Would you consider participating in research remotely (telephone/online)?' Of the 163 people who answered the question, 145 (89%) indicated that they would be happy to take part in remote research. Based on this feedback, we submitted an amendment to our existing research ethics that was approved in October 2020. Following this, we conducted a pilot remote testing session with an older healthy control individual who was also a carer for a family member living with dementia. Their feedback was instrumental in developing and improving our remote testing procedure.

Results from this work will be disseminated to members of the support groups that we run with Rare Dementia Support (www.raredementiasupport.org) through online presentations at webinars and research summaries in newsletters.

### RESULTS

General characteristics of participant groups are presented in table 1; performance on the general neuropsychological tests is given in table 2; performance on the neurolinguistic tests is shown in table 3. Figures 1 and 2 show radar plots of performance for each participant group for the general neuropsychological and neurolinguistic tests, respectively. Figures 3 and 4 show performance profiles of healthy control participants on the general neuropsychological and neurolinguistic tests, respectively. online supplemental figure 2 and 3 show performance profiles of the combined patient cohort on the general neuropsychological and neurolinguistic tests, respectively. Bayesian statistics are presented in online supplemental tables 2 and 3; equality of variance analyses is presented in online supplemental table 4; and results for the audibility screening task (see online supplemental methods for more information) are presented in online supplemental table 5.

### General participant characteristics

Of the 87 potential participants who were contacted, 35 (40.2%) ultimately took part in the research (25 patients, 10 healthy controls who had taken part in face-to-face research previously). Reasons for declining participation and reports of any technical issues with remote test delivery are detailed in the online supplemental material.

There were no significant differences in age, years of education or symptom duration between the face-to-face and remote testing patient cohorts (table 1).

Below we highlight comparisons where there was substantial evidence in support of either the null (ie, no difference between remote and face-to-face performance) or alternative (ie, difference between remote and face-to-face performance) hypothesis. Comparisons are shown in full in online supplemental tables 2 and 3.

### General neuropsychological assessment

Overall, there was little evidence for a significant effect of assessment environment on general neuropsychological test performance in any participant group.

Healthy individuals scored equally well on the digit span reverse, the British Picture Vocabulary Scale (BPVS), the Graded Difficulty Arithmetic test (GDA), and on both letter and category fluency tests (all $BF_{01}$ >3 indicating substantial evidence in favour of the null hypothesis). However, they performed less well on the Visual Object and Spatial Perception object decision task (VOSP) ($BF_{01}$=0.0404, indicating substantial evidence in favour of the alternative hypothesis) in remote testing than in face-to-face testing, with the remote group (mean=18.4) performing worse than the face-to-face group (mean=19.5) (figures 1 and 3, table 2 and online supplemental table 2).

For the comparisons of the combined remote versus combined face-to-face patient cohorts, there was substantial evidence supporting the null hypothesis for all neuropsychological tests (all $BF_{01}$ >3), except for the National Adult Reading Test and both letter and category fluency tests, where evidence in support of the null hypothesis was anecdotal (table 2 and online supplemental table 2).

For individual patient groups (figure 1, table 2 and online supplemental table 2), there was substantial evidence to suggest that the remote AD cohort performed similarly to the face-to-face AD cohort on Wechsler Adult Intelligence Scale (WASI) matrix reasoning, digit span

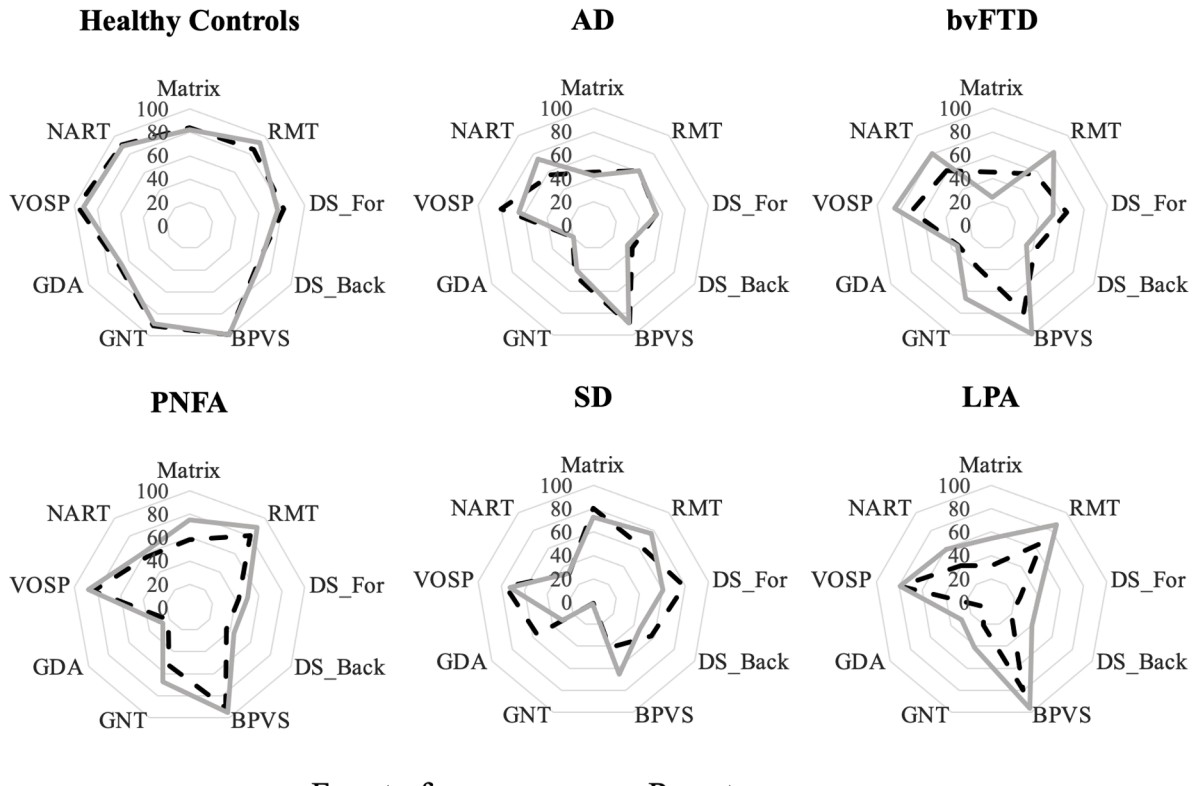

**Figure 1** Radar plots of general neuropsychological test performance, by participant group and testing environment. Average percentage correct score (plotted on concentric lines) was calculated for each participant group for each test in the neuropsychological tests, across each testing environment. Scores for the fluency tasks were not included here as responses on these tasks cannot be evaluated as correct/incorrect. AD, patient group with typical Alzheimer's disease; BPVS, British Picture Vocabulary Scale; bvFTD, patient group with behavioural variant frontotemporal dementia; DS_For/Back, digit span forwards/backwards; GDA, Graded Difficulty Arithmetic test; GNT, Graded Naming Test; LPA, patient group with logopenic progressive aphasia; Matrix, WASI matrix reasoning; NART, National Adult Reading Test; PNFA, patient group with progressive non-fluent aphasia; RMT, Recognition Memory Test; SD, patient group with semantic dementia; VOSP, Visual Object Space Perception; WASI, Wechsler Adult Intelligence Scale.

forwards, Graded Naming Test (GNT), BPVS, GDA and category fluency test (all $BF_{01}$ >3). However, the remote AD cohort performed less well on the VOSP (mean=13.0; $BF_{01}$=0.171, substantial evidence) compared with the face-to-face cohort (mean=16.1). Conversely, patients with LPA who completed the letter fluency test remotely (mean words=11.3) performed better than those who completed the same task face-to-face (mean=2.6; $BF_{01}$=0.188, substantial evidence).

No other comparisons yielded substantial evidence in support of either hypothesis (online supplemental table 2).

Results of the equality of variance analyses are reported in full in online supplemental table 4. The assumption of homogeneity of variance was not violated for any test in the general neuropsychological tests in either the healthy control or combined patient cohorts.

## Neurolinguistic assessment

Overall, there was little evidence for a significant effect of assessment environment on neurolinguistic test performance in any participant group.

Healthy individuals scored equally well on the Boston Naming Test (BNT), the camel and cactus test and the bisyllabic single-word repetition test (all $BF_{01}$ >3, indicating substantial evidence in favour of the null hypothesis). However, they performed less well on the monosyllabic word repetition test ($BF_{01}$=0.0487, substantial evidence in favour of the alternative hypothesis) in remote testing than in face-to-face testing, with the remote group (mean=12.7) performing worse than the face-to-face group (mean=14.6) (figures 2 and 4, table 3 and online supplemental table 3).

The comparisons between combined patient cohorts for remote versus face-to-face testing showed substantial evidence supporting the null hypothesis for non-word reading, concrete synonyms, the Psycholinguistic Assessments of Language Processing in Aphasia-55, and bisyllabic and trisyllabic single-word repetition tests (all $BF_{01}$ values >3). There was anecdotal evidence supporting the null hypothesis on all other neurolinguistic tests (all $BF_{01}$ values between 1 and 3).

**Figure 2** Radar plots of performance on neurolinguistic test performance, by participant group and testing environment. Average percentage score (plotted on concentric lines) was calculated for each participant group for each test in the neurolinguistic tests, across each testing environment. Abstract, abstract synonyms test; bi rep, bisyllabic single-word repetition; BNT, Boston Naming Test; C & C, camel and cactus test; concrete, concrete synonyms test; irregular, irregular word reading test; LPA, patient group with logopenic progressive aphasia; mono rep, monosyllabic single-word repetition test; non word, non-word reading test; PALPA, Psycholinguistic Assessment of Language Processing in Aphasia subtests; PNFA, patient group with progressive non-fluent aphasia; regular, regular word reading test; SD, patient group with semantic dementia; sentence rep, graded difficulty sentence repetition test; spoken sentences, spoken sentences test; tri rep, trisyllabic single-word repetition test.

Individual patient group comparisons across environments did not yield substantial evidence in support of either hypothesis.

Results of the equality of variance analyses are reported in full in online supplemental table 4. For the healthy control group, the assumption of equality of variance was violated for monosyllabic repetition ($F=0.11$, $p<0.05$, with higher variability in the remote condition); for the combined patient cohort, this assumption was violated for monosyllabic repetition ($F=4.89$, $p=0.03$, with higher variability in the remote group), the camel and cactus test ($F=4.25$, $p=0.02$, with higher variability in the face-to-face group) and the spoken sentence task ($F=4.37$, $p<0.05$, with higher variability in the face-to-face group).

### Feedback on remote testing experience

We received 20 responses to the question, 'How comfortable did you feel in this new setting (eg, online testing)?' after the general neuropsychology remote session (from 10 healthy control participants and 10 patients); and 22 responses to the same question that was posed after the neurolinguistic remote session (from 10 healthy control participants and 12 patients). There was little evidence for a significant difference across groups on these ratings for either the general neuropsychology session (control mean=9.60, standard deviation (st.d)=0.70; patient mean=9.10, st.d=1.91; $BF_{01}=1.825$, anecdotal evidence supporting the null hypothesis) or neurolinguistic session (control mean=9.10, st.d=1.73; patient mean=8.92, st.d=1.73; $BF_{01}=2.325$, anecdotal evidence supporting the null hypothesis).

### DISCUSSION

The present findings suggest that administration of neuropsychological tasks remotely over the internet with healthy older adults and people with a diverse range of dementia phenotypes is feasible according to three key metrics: acceptability, adaptation and demand.[11] In terms

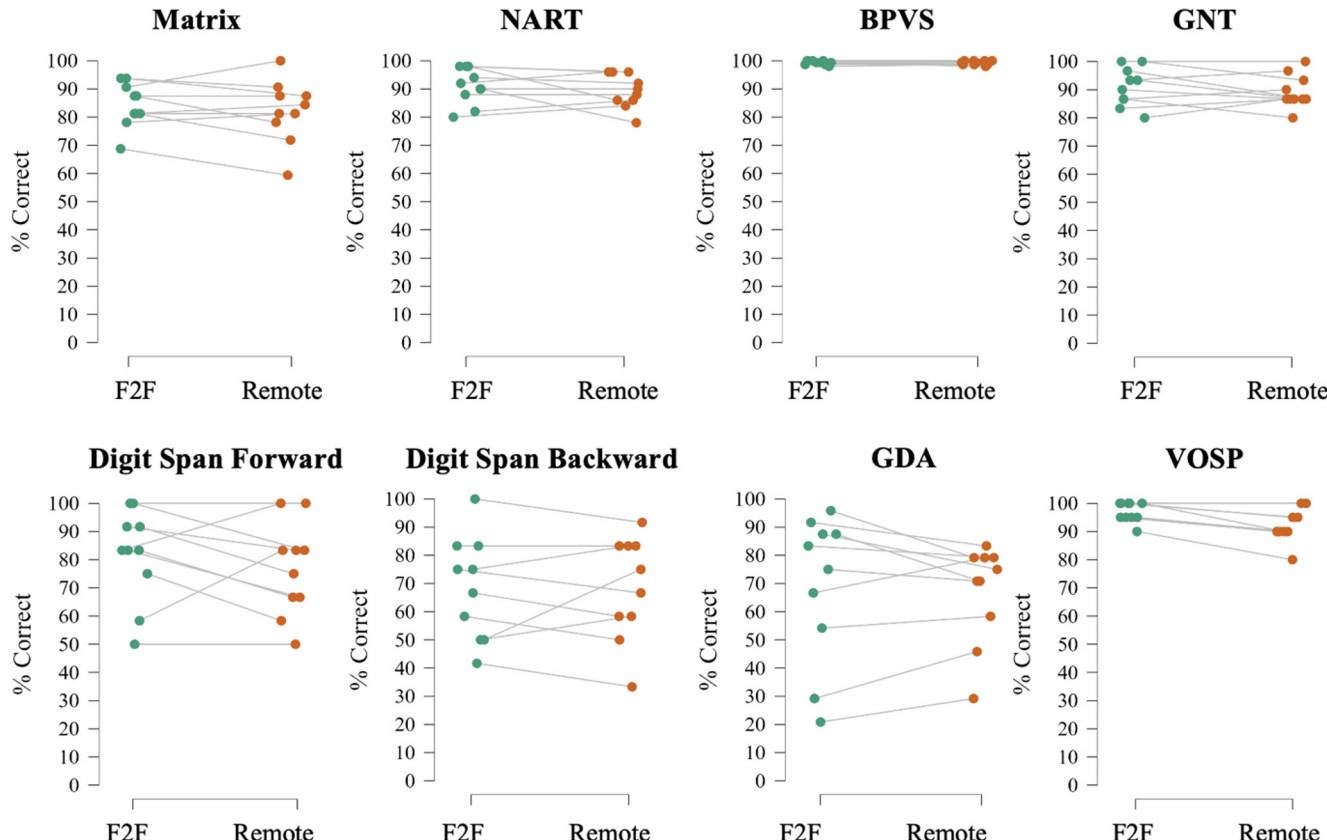

**Figure 3** Performance profiles of healthy control participants on tasks in general neuropsychological tests. Line plots showing performance profiles of individual healthy control participants on tasks in the general neuropsychological tests. BPVS, British Picture Vocabulary Scale; F2F, face-to-face; GDA, Graded Difficulty Arithmetic test; GNT, Graded Naming Test; Matrix, WASI matrix reasoning; NART, National Adult Reading Test; VOSP, Visual Object Space Perception object decision task; WASI, Wechsler Adult Intelligence Scale.

of acceptability, results from our feedback questionnaires indicated that patient and healthy control participants were comfortable with the remote testing environment. For adaptation, we have demonstrated that similar performance outcomes were obtained across test settings (with further discussion of these findings below). In terms of demand, only 6.9% of those we contacted about taking part in the research declined due to technological reasons (online supplemental material 1).

Our Bayesian analytical approach demonstrated that there was anecdotal or substantial evidence suggesting comparable performance across testing environments of healthy participants and patients with AD on a range of general neuropsychological and neurolinguistic tests, specifically those targeting working memory (digit span forward), executive functioning (digit span reverse, letter and category fluency tests, WASI matrix reasoning), arithmetic skills (GDA) and general semantic knowledge (BNT, BPVS, GNT). These results corroborate previous reports of preserved neuropsychological performance on executive function, working memory, and language tests across testing environments in both healthy individuals and patients with AD.[10] Our findings also corroborated previous work suggesting that remote assessments are viable for people with PPA.[23]

Healthy controls and participants with AD both performed significantly worse on the remote version of the VOSP object decision task, in which participants are presented with four silhouettes and asked to select the drawing of a real object; the three distractor silhouettes are based on nonsense shapes. The typical amnestic AD phenotype can include prominent visuospatial impairments[24 25] and it is feasible that a reduction in stimulus quality may have stressed cortical apperceptive mechanisms still further, akin to a dynamic 'stress test' of degraded input processing.[26–28] However, it is worth noting that there was no such discrepancy across the AD cohorts for other tasks involving visual administration (eg, WASI matrix). For the healthy controls, the absolute performance difference across environments was relatively small (mean reduction of 1.1 points) for the VOSP; however, we note that even small differences can have important consequences if this change were to yield a lower scaled/percentile score and thus affect interpretations of test performance. It is also possible that this reduction at least in part reflected normal healthy ageing, consistent with previous findings,[29] as the healthy control cohort was tested on the remote test 3–4 years after their face-to-face assessment.

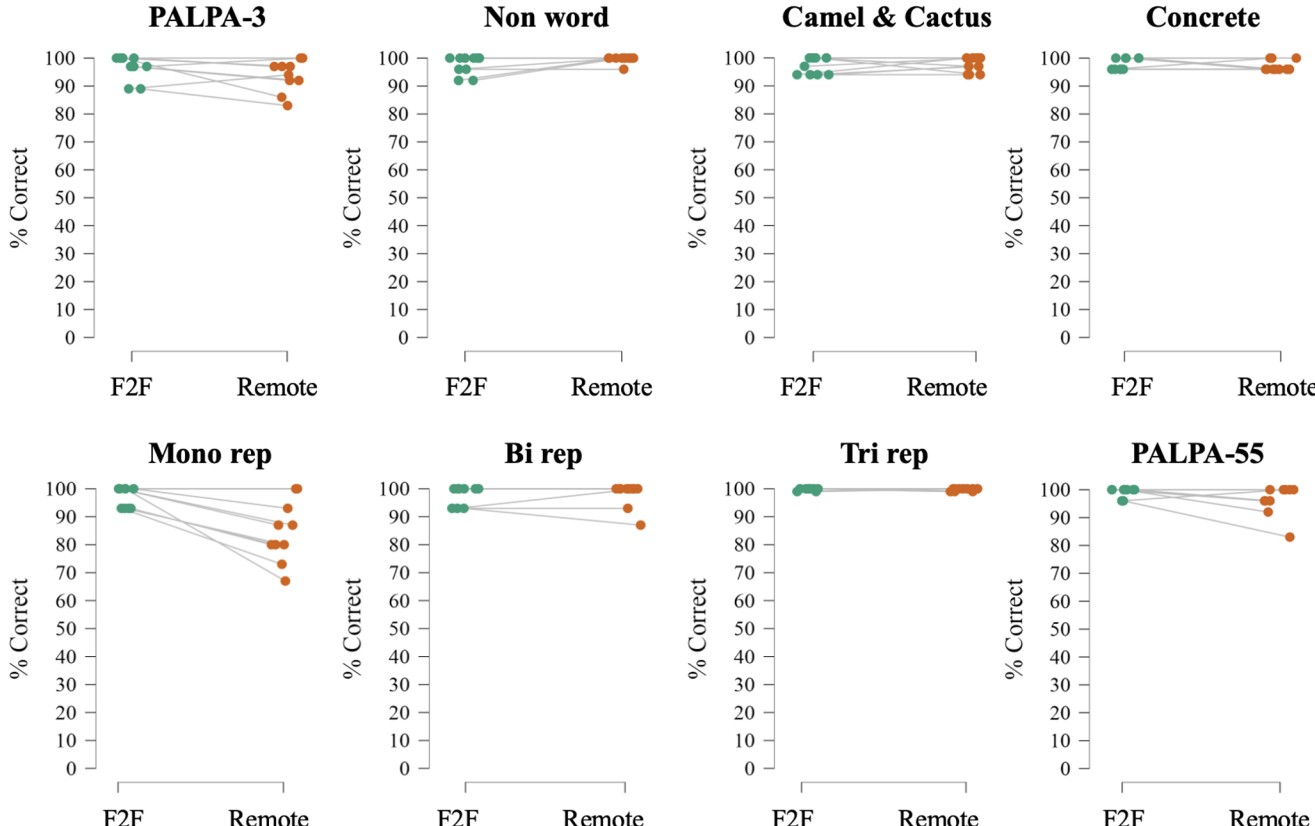

**Figure 4** Performance profiles of healthy control participants on tasks in the neurolinguistic tests. Line plots indicating percentage scores for each healthy control on representative tests from the neurolinguistic tests administered face-to-face (F2F) and remotely. Scores on the trisyllabic single-word repetition task were jittered slightly on the x-axis to allow for plotting as participants were uniformly at ceiling in both environments. Bi rep, bisyllabic single-word repetition; Concrete, concrete synonyms test; Mono rep, monosyllabic single-word repetition test; Non word, non-word reading test; PALPA, Psycholinguistic Assessment of Language Processing in Aphasia; Tri rep, trisyllabic single-word repetition.

Healthy control participants also performed significantly worse on the monosyllabic single-word repetition task when delivered remotely, and there was more variability on this task when performed remotely. The videoconferencing software may have degraded the fidelity of the raw speech signal,[30] essentially resulting in a harder task than when administered face-to-face. This is potentially consistent with the controls' preserved performance on the bisyllabic single-word repetition test where top-down information can be used to complement bottom-up auditory information partially degraded by the videoconferencing software.[31] An alternative (or complementary) explanation could again be age-related changes, here affecting hearing function (presbycusis).[32–34]

The finding of significantly better performance on the verbal (letter) fluency task in the LPA cohort tested remotely compared with the cohort tested face-to-face is surprising. The obvious explanation is that the remote cohort was overall less impaired than the patients seen face-to-face; although efforts were made to match the two cohorts for disease severity and other potentially relevant factors. An alternative explanation could be that participants found the remote setting less anxiety provoking than face-to-face testing in an unfamiliar environment. Patients with LPA may be relatively susceptible to anxiety

as a factor modulating cognitive performance.[35 36] Additionally, as word retrieval is an intrinsically dynamic process that is likely to be facilitated by the availability of 'prompts',[37] patients may have benefitted from cueing of word retrieval by their more familiar home environments.

The way the neuropsychological and neurolinguistic testing protocol was adapted for remote delivery may have favoured null differences. The testing sessions were shorter and spread out within a week, which may have helped counteract the effect of anxiety related to the unfamiliarity of the remote testing setting, as well as potential 'Zoom' fatigue.[17] The increased flexibility of scheduling compared with face-to-face testing in addition to the absence of potential stressors associated with a face-to-face research visit (eg, travelling, being in a unfamiliar environment) may have led to participants feeling more relaxed when taking part remotely versus face-to-face: future research should explore this. Furthermore, certain tests selected for remote delivery may have been intrinsically less susceptible to changes in testing protocol (eg, BPVS), whereas we deliberately excluded tests that we considered would not be practical or suboptimal for remote delivery (eg, WASI block design, Baxter spelling test, Trails). Anecdotally, participants reported satisfaction with the remote testing protocol.

The current study presents several limitations which should inform future work. First, while most statistical comparisons indicated similar performance between testing environments for healthy participants and those with dementia, they were not all supported by substantial evidence and certain comparisons even led to the opposite conclusion. Second, the present study was not ideally designed to compare the two testing environments, as the patient cohorts were different and the healthy control participants were not tested simultaneously in both environments within the same year. This meant that certain statistical measures that may have been informative (eg, assessing stability of ranking within groups) were not appropriate here. While it is likely that the variability of test results in the patient cohorts will be influenced by testing environment, this needs to be interpreted cautiously, due to unequal sample sizes and individual disease trajectories—and may further depend on the particular test employed. Indeed, our sample sizes across modalities were relatively small. These findings would need to be replicated in larger cohorts with the same patients in each test situation, to rule out the possibility of small differences observed in favour of face-to-face testing—and in particular, to assess the extent of individual variability in any differential effect of test environment. Patients of equivalent disease severity would also need to be tested to compare the differential impact of diagnosis on remote performance over the course of the illness. Third, here we did not control for potential deficits in peripheral hearing as these are difficult to measure remotely without adequate equipment. Fourth, we manually adapted face-to-face tasks for remote administration, but there are now several established fully integrated online neuropsychological tests that have shown success in assessing patients with neurodegenerative disease remotely[38]: future research could explore the extent to which our results are comparable with those obtained by such tests. Fifth, only a proportion of patients had time to respond to the additional question asking for their rating of comfort with the remote testing set-up, and it is possible that there was some selection bias here in that those patients who felt most comfortable with the technology finished the sessions earlier and therefore had time to give this additional feedback. Relatedly, while there was little evidence for a significant effect of 'group' (controls vs patients) on these comfort ratings, the patient cohort did record a lower mean average score than controls after both the remote general neuropsychological and neurolinguistic sessions, something that would warrant further and more in-depth investigation with validated assessment tools. Finally, we note that the study was designed to explore the potential for remote neuropsychological assessments of research participants, and the results and conclusions here may not generalise to clinical settings.

Overall, the present findings demonstrate that, despite challenges in setting up remote testing protocols (specifically due to technological requirements), these may produce similar results to face-to-face testing protocols. These are encouraging findings given the current climate and anticipating that research participants may continue to favour remote (or hybrid) visits over face-to-face assessments for reasons of convenience as well as safety, as we move beyond the COVID-19 pandemic.

**Author affiliations**
[1]Dementia Research Centre, University College London, London, UK
[2]Kidney Cancer Program, UT Southwestern Medical Center, Dallas, Texas, USA
[3]Cognition and Brain Sciences Unit, Cambridge University, Cambridge, UK

**Acknowledgements** We are grateful to all participants for their involvement.

**Contributors** M-CR-K, JJ, LD, EB, LR, RLB, EVB, CG, SB, JR, SC, JW and CH contributed to the conception and design of the study. M-CR-K, JJ, LD, EB, LR, RLB, EVB, CG and CH contributed to the acquisition of data. M-CR-K, JJ, JW and CH were involved in the analysis and interpretation of data, and wrote the first draft of the article. M-CR-K, JJ, LD, EB, LR, RLB, EVB, CG, SB, JR, SC, JW and CH were involved in revision of the draft and approved the final version of the manuscript. CH is responsible for the overall content as the guarantor. M-CR-K and JJ contributed equally to this paper.

**Funding** The Dementia Research Centre is supported by Alzheimer's Research UK, Brain Research UK and the Wolfson Foundation. This work was supported by the Alzheimer's Society, the Royal National Institute for Deaf People, Alzheimer's Research UK, the National Institute for Health Research University College London Hospitals Biomedical Research Centre and the University College London Leonard Wolfson Experimental Neurology Centre (grant PR/ylr/18575). M-CR-K was supported by a Wellcome Trust PhD Studentship (102129/B/13/Z). JJ is supported by a Frontotemporal Dementia Research Studentship in Memory of David Blechner (funded through the National Brain Appeal). EB was supported by a Brain Research UK PhD Studentship. RLB was supported by an MRC PhD Studentship in Mental Health. SC was supported by grants from ESRC-NIHR (ES/L001810/1), EPSRC (EP/M006093/1) and Wellcome Trust (200783). CH was supported by a Royal National Institute for Deaf People–Dunhill Medical Trust Pauline Ashley Fellowship (grant PA23_Hardy) and a Wellcome Institutional Strategic Support Fund Award (204841/Z/16/Z). This research was funded in part by UKRI and the Wellcome Trust (grant 204841/Z/16/Z). For the purpose of Open Access, the author has applied a Creative Commons Attribution (CC BY) public copyright licence to any Author Accepted Manuscript version arising from this submission.

**Competing interests** None declared.

**Patient and public involvement** Patients and/or the public were involved in the design, or conduct, or reporting, or dissemination plans of this research. Refer to the Methods section for further details.

**Patient consent for publication** Not required.

**Ethics approval** This study involves human participants and ethical approval was granted by the University College London and National Hospital for Neurology and Neurosurgery Joint Research Ethics Committees in accordance with the Declaration of Helsinki (reference numbers 06N032 and 150508). Participants gave informed consent to participate in the study before taking part.

**Provenance and peer review** Not commissioned; externally peer reviewed.

**Data availability statement** Data are available upon reasonable request. The data that support the findings of this study are available on request from the corresponding author.

terminology, drug names and drug dosages), and is not responsible for any error and/or omissions arising from translation and adaptation or otherwise.

**ORCID iDs**
Maï-Carmen Requena-Komuro http://orcid.org/0000-0002-5624-0527
Jonathan D Rohrer http://orcid.org/0000-0002-6155-8417
Jason D Warren http://orcid.org/0000-0002-5405-0826
Chris JD Hardy http://orcid.org/0000-0002-4900-6492

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
