## [Reviewer comments · BMJ Open]

ARTICLE DETAILS

TITLE (PROVISIONAL)	Remote versus face-to-face neuropsychological testing for dementia research: a comparative study in patients with Alzheimer's disease, patients with frontotemporal dementia, and healthy older individuals
AUTHORS	Requena-Komuro, Mai-Carmen; Jiang, Jessica; Dobson, Lucianne; Benhamou, Elia; Russell, Lucy; Bond, Rebecca L; Brotherhood, Emilie V.; Greaves, Caroline; Barker, Suzie; Rohrer, Jonathan; Crutch, Sebastian; Warren, Jason; Hardy, Chris

VERSION 1 – REVIEW

REVIEWER	Huygelier, Hanne KU Leuven
REVIEW RETURNED	30-Jun-2022

GENERAL COMMENTS	In this paper the authors examined the extent to which neuropsychological assessment was feasible through videocall software and whether the neuropsychological testing through videocall produces equivalent results relative to face-to-face testing. In general the paper is clearly written and touches upon an important issue. However, there are some issues with the methodology that require careful consideration. Two key aspects that could be improved upon are the conceptualization and operationalization of "feasibility" and the methods used to demonstrate the equivalence of remote vs face-to-face testing. I will elaborate on these (and other) concerns / suggestions to improve the paper in the points below. 1) The authors aimed to examine the feasibility of remote testing in a home environment. However, no definition of feasibility is provided and the operationalization of this concept is unclear. Does feasibility refer to the percentage of patients that can be reached with remote testing, the percentage of incomplete test batteries, the experience of the patients / caregivers with regard to the testing, the number of tests one can adapt to the videocall platform, the extent to which remote vs face-to-face testing is equivalent, the cost of performing such an assessment, the time a clinician has to invest in the testing and preparing the testing, the content validity of the remote test battery or several of these and other criteria? What criteria do you use to judge whether something is feasible or not? In addition, in the discussion the authors conclude that remote testing was "broadly feasible". What criteria did you use to obtain this conclusion? 2) The authors assessed whether there were differences in average performance on the neuropsychological tests. However they do not explain why equivalence between remote and face-to-face testing is important to assess. Moreover, to demonstrate that remote and face-to-face testing are equivalent to each other, only testing for differences in average performance is not sufficient. That is, the similarity in average performance can indicate that there is no systematic bias in performance based on the remote testing relative
---

to the face-to-face testing, but it does not demonstrate that the remote testing is a valid assessment necessarily. That is, in addition to differences in means, variability in performance may increase in remote testing due to some patients experiencing more struggles with the videocall software than others (see for instance the results on the neurolinguistic test mono rep in Figure 4). I suggest that the authors add a test of equality of variance. Last, the stability of ranking of participants should in an ideal scenario also be assessed. In theory, you could have the exact same average and standard deviation (suggesting equivalence), but an association of -1 (signaling a lack of equivalence). Given the long test-retest interval for healthy controls and between-subject design for patients, looking at stability of ranking is not very informative, but this could be elaborated on in the discussion.

3) The results of healthy controls are visualized in Figure 3 and Figure 4 such that readers can assess the variability and type of score distribution. However, no such visualization of the patient data is provided. Can the authors add a figure showing the distribution of test scores in both conditions for the whole patient sample? This would give the reader increased insight in the results.

4) The authors used a BF threshold of 3 and 1/3 to infer “strong evidence” either in favor of H1 or H0 and label results in between those values as “anecdotal evidence”. In most cases, a BF threshold of 10 or 1/10 is used to conclude “strong evidence”, while 3 and 1/3 is interpreted as “substantial evidence”. (See for instance: (Kass & Raftery, 1995)). I highly recommend to change the labels associated to the BFs in this paper in line with these recommendations to avoid that readers overinterpret the strength of evidence.

5) For some tests (e.g., VOSP) there was a difference between remote and face-to-face testing in average performance. The authors then indicate that this absolute difference in performance was small (18.4 vs 19.5). A small difference can however in certain situations have a large impact. For instance, if a cut-off is used to interpret the test scores as “impaired” vs “non-impaired” one may fall into a different category by scoring one point lower or higher. So interpreting these raw score differences as small requires more substantiation. Could such a difference of 1 point affect scaled scores of patients or interpretations of the results on the test?

6) The authors report the average performance on several tests in Table 2 for the healthy controls and the separate patient groups, but not for the patient group as a whole. In supplementary materials they report the Bayes Factors for all group comparisons. The comparisons for the separate patient groups are almost always inconclusive (BF in between 1/3 and 3 as reported in the Supplementary Materials) while the comparisons on the entire patient sample did provide some evidence either in favor of H0 or H1. Given those results, I would report the average and SD of the tests for the healthy control and total patient sample in the main text and include the BFs in that table as those are the most informative results and report the stats for the separate patient groups all in supplementary as they are not very informative (inconclusive results).

7) Some details that could be further clarified:

- a. The authors reported that 6 individuals declined to participate in the study due to not being comfortable with the videocall technology. It would be good to know how many participants were invited to take part in the study to make this number more interpretable.
- b. An initial telephone screen was conducted to evaluate whether participants had access to the necessary equipment, a quiet testing place and no severe visual or auditory impairments. How many

	potential participants were excluded in this phase and what were the most common reasons?
--	---

REVIEWER	Chaytor, Naomi Washington State University
REVIEW RETURNED	21-Aug-2022

GENERAL COMMENTS	The focus of this paper is on comparison between TeleNeuropsychology (to patient's homes) and face-to-face neuropsychological testing in older adults with and without various forms of dementia. The manuscript is well-written and the topic is of high interest. However, I have several major concerns that limit my enthusiasm. These are outlined below: Abstract: The results section does not report any data – statistics should be reported to support the conclusions Introduction: Well written and concise Methods: More detail is needed regarding how patients were diagnosed, both the remote cohort and the face-to-face cohort. Was the diagnosis based on clinical presentation/neurological work-up? Was any of the neuropsychological data collected used for diagnostic purposes? The authors state that a T-MMSE of 12 was used as an inclusion criteria – does this mean they had to perform above or below this threshold to be included? The test battery is very sparse in some areas and has some odd choices for episodic memory (why no verbal memory?), working memory (digit span forward is not working memory, as there is no manipulation of information) and executive functioning (digit span backward is more accurately conceptualized as working memory, while semantic fluency is a language task). More detail is needed regarding the remote testing conditions. How many used a tablet vs laptop vs desktop computer? How many had someone else present in the room? What were the instructions for the remote test environment? Closed door? Pets? No sound? Turn off phones/devices? Did they have to show the examiner the environment to make sure there was no paper or calculator (for example)? Did distractions occur (e.g., interruptions from family, phone ringing?) Discussion: A major threat to the generalizability of these results to clinical settings is that participants were recruited to a remote testing study rather than evaluating clinical participants who needed to be remote during covid to determine how these procedures would work in those who are not carefully selected/self-selected.
--

VERSION 1 – AUTHOR RESPONSE

Reviewer 1 (Dr. Hanne Huygelier)

We thank Dr. Huygelier for their helpful comments and have addressed these important points below.

- 1. The authors aimed to examine the feasibility of remote testing in a home environment. However, no definition of feasibility is provided and the operationalization of this**

concept is unclear. Does feasibility refer to the percentage of patients that can be reached with remote testing, the percentage of incomplete test batteries, the experience of the patients / caregivers with regard to the testing, the number of tests one can adapt to the videocall platform, the extent to which remote vs face-to-face testing is equivalent, the cost of performing such an assessment, the time a clinician has to invest in the testing and preparing the testing, the content validity of the remote test battery or several of these and other criteria? What criteria do you use to judge whether something is feasible or not? In addition, in the discussion the authors conclude that remote testing was “broadly feasible”. What criteria did you use to obtain this conclusion?

Whilst a formal feasibility study was beyond the scope of our research, we appreciate that the term ‘feasibility’ has very specific connotations and apologise for not being clearer about this previously. We appreciate having the opportunity to clarify our definition of ‘feasible’ and have done this in our Introduction:

There are three main considerations that need addressing: (1) Adaptation: are similar performance outcomes obtained on neuropsychological tests designed for face-to-face administration when given remotely? If performance across modalities is equivalent then this could allow for pooling of data collected face-to-face and remotely, potentially allowing more equitable access to research for participants who are not able to physically travel to research centres. (2) Demand: to what extent are research participants with and without dementia willing to engage in remote neuropsychological research? And (3) Acceptability: how satisfactory is remote testing for research participants including those with diverse forms of dementia?”. (page 3, lines 33-40).

Additionally, we now report the percentage of people who were contacted about the remote research who agreed to participate in our Results (page 13, lines 3-6) to address the ‘Demand’ criterion (reasons for not participating are now given in Supplementary Material, page 1-2, lines 40-52). We have also added information about a question pertaining to participants’ level of comfort with the remote testing environment that were administered to assess the ‘Acceptability’ criterion in the Methods (page 6, lines 46-48; page 7, lines 35-38) and Results (page 14, lines 32-43).

We also appreciate that the term ‘broadly feasible’ was confusing and unhelpful, and have changed this to ‘feasible’, with the following discussion:

“The present findings suggest that administration of neuropsychological tasks remotely over the internet with healthy older adults and people with a diverse range of dementia phenotypes is feasible according to three key metrics: acceptability, adaptation, and demand¹¹. In terms of acceptability, results from our feedback questionnaires indicated that patient and healthy control participants were comfortable with the remote testing environment. For adaptation, we have demonstrated that similar performance outcomes were obtained across test settings (with further discussion of these findings below). And in terms of demand, only 6.9% of those we contacted about taking part in the research declined due to technological reasons (Supplementary Materials online). (page 14, lines 46-48; page 15, lines 1-6)

- 2. The authors assessed whether there were differences in average performance on the neuropsychological tests. However they do not explain why equivalence between remote and face-to-face testing is important to assess. Moreover, to demonstrate that remote and face-to-face testing are equivalent to each other, only testing for differences in average performance is not sufficient. That is, the similarity in average performance can indicate that there is no systematic bias in performance based on the remote testing relative to the face-to-face testing, but it does not demonstrate that the remote testing is a valid assessment necessarily. That is, in addition to differences in means, variability in performance may increase in remote testing due to some patients experiencing more struggles with the videocall software than others (see for instance the results on the neurolinguistic test mono rep in Figure 4. I suggest that the authors add a test of equality of variance. Last, the stability of ranking of participants should in an ideal scenario also be assessed. In theory, you could have the exact same average and standard deviation (suggesting equivalence), but an association of -1 (signaling a lack of equivalence). Given the long test-retest interval for healthy controls and**

between-subject design for patients, looking at stability of ranking is not very informative, but this could be elaborated on in the discussion.

We have added an explanation as to why it is important to assess equivalence between remote and face-to-face testing - this is incorporated in our response to Point 1 above.

In terms of variability in performance across modalities, we have added a new Figure S2 and Figure S3 in response to Dr. Huygelier's point below. This shows the spread of scores across the combined patient sample on a range of the neuropsychological measures used (chosen to match those included in the pre-existing Figures 3 and 4). We agree that it is very difficult to draw firm conclusions about variability in between-subject designs, particularly with small sample sizes as featured here. Nevertheless, we have conducted tests of equality of variance as requested. These are described in the Methods (page 7, lines 40-41), reported in the Results (page 13, lines 47-49, page 14, lines 24-30), and reported fully in new Table S4 in Supplementary Materials.

We agree that tests assessing stability of ranking over time would have been helpful and informative with a different experimental design. We have taken Dr Huygelier's helpful suggestion on board and taken the opportunity to elaborate on this in the Discussion (page 16, lines 21-28):

"Second, the present study was not ideally designed to compare the two testing environments, as the patient cohorts were different and the healthy control participants were not tested simultaneously in both environments within the same year. This meant that certain statistical measures that may have been informative (e.g. assessing stability of ranking within groups) were not appropriate here. While it is likely that the variability of test results in the patient cohorts will be influenced by testing environment, this needs to be interpreted cautiously, due to unequal sample size and individual disease trajectories - and may further depend on the particular test employed."

- 3. The results of healthy controls are visualized in Figure 3 and Figure 4 such that readers can assess the variability and type of score distribution. However, no such visualization of the patient data is provided. Can the authors add a figure showing the distribution of test scores in both conditions for the whole patient sample? This would give the reader increased insight in the results.**

Please see our response to Point 3, above.

- 4. The authors used a BF threshold of 3 and 1/3 to infer "strong evidence" either in favor of H1 or H0 and label results in between those values as "anecdotal evidence". In most cases, a BF threshold of 10 or 1/10 is used to conclude "strong evidence", while 3 and 1/3 is interpreted as "substantial evidence". (See for instance: (Kass & Raftery, 1995)). I highly recommend to change the labels associated to the BFs in this paper in line with these recommendations to avoid that readers overinterpret the strength of evidence.**

We have adopted the labels associated with the BFs that Dr. Huygelier has kindly suggested. We have made the appropriate changes in the Methods (page 7, line 30), Results (pages 13-14), Discussion (pages 15-16), Table 2 legend (page 9), Table 3 legend (page 11), Table S2 (Supplementary material, page 5), Table S3 (Supplementary material, page 6).

- 5. For some tests (e.g., VOSP) there was a difference between remote and face-to-face testing in average performance. The authors then indicate that this absolute difference in performance was small (18.4 vs 19.5). A small difference can however in certain situations have a large impact. For instance, if a cut-off is used to interpret the test scores as "impaired" vs "non-impaired" one may fall into a different category by scoring one point lower or higher. So interpreting these raw score differences as small requires more substantiation. Could such a difference of 1 point affect scaled scores of patients or interpretations of the results on the test?**

We agree with this excellent point and have now elaborated on this in the Discussion:

“For the healthy controls, the absolute performance difference across environments was relatively small (mean reduction of 1.1 points) for the VOSP; however we note that even small differences can have important consequences if this change were to yield a lower scaled/percentile score and thus affect interpretations of test performance. It is also possible that this reduction at least in part reflected normal healthy ageing, consistent with previous findings⁴⁸, as the healthy control cohort was tested on the remote battery three-to-four years after their face-to-face assessment.” (page 15, lines 26-32).

6. **The authors report the average performance on several tests in Table 2 for the healthy controls and the separate patient groups, but not for the patient group as a whole. In supplementary materials they report the Bayes Factors for all group comparisons. The comparisons for the separate patient groups are almost always inconclusive (BF in between 1/3 and 3 as reported in the Supplementary Materials) while the comparisons on the entire patient sample did provide some evidence either in favor of H0 or H1. Given those results, I would report the average and SD of the tests for the healthy control and total patient sample in the main text and include the BFs in that table as those are the most informative results and report the stats for the separate patient groups all in supplementary as they are not very informative (inconclusive results).**

We appreciate this suggestion. We are keen to retain the separate patient group performance in the main Tables given the different clinical profiles across each group. However, we have now added new columns to Tables 2 and 3 that display the data Dr Huygelier has requested for the combined patient groups.

87 Some details that could be further clarified:

- a. **The authors reported that 6 individuals declined to participate in the study due to not being comfortable with the videocall technology. It would be good to know how many participants were invited to take part in the study to make this number more interpretable.**

We have now added a full description of how many potential participants were contacted, and the reasons they were not included in the study (Supplementary Material, pages 1-2, lines 40-52).

- b. **An initial telephone screen was conducted to evaluate whether participants had access to the necessary equipment, a quiet testing place and no severe visual or auditory impairments. How many potential participants were excluded in this phase and what were the most common reasons?**

None were excluded at this stage and we have now added a line to address this point (page 4, line 30).

Reviewer 2 (Dr. Naomi Chaytor)

We thank Dr. Chaytor for their helpful comments and have addressed these important points below.

1. **Abstract: The results section does not report any data – statistics should be reported to support the conclusions**

Please see our response to the Editors' comments above (Point 2).

2. **More detail is needed regarding how patients were diagnosed, both the remote cohort and the face-to-face cohort. Was the diagnosis based on clinical presentation/neurological work-up? Was any of the neuropsychological data collected used for diagnostic purposes?**

We have specified the routes via which patients were referred to us for research in the Methods (page 4, lines 17-22). As we are reporting on a research (rather than clinical) cohort we are unable to give detailed information about the clinical diagnostic procedure, which in any case would vary depending on the clinician and clinic in which each patient was seen. We did, however, ensure that all patients met relevant research consensus criteria and have included a detailed description of how patients

were evaluated against these consensus criteria for the purposes of our research (page 4, lines 32-46). We have also added a sentence to the Methods specifying that there was no circularity in terms of the neuropsychological data reported in this research study: “The neuropsychological tests reported in this research article were not used for diagnostic purposes” (page 4, lines 43-44).

3. The authors state that a T-MMSE of 12 was used as an inclusion criteria – does this mean they had to perform above or below this threshold to be included?

Participants had to perform at or above this threshold to be included. We have clarified this in the Methods by replacing the word ‘cut-off’ with ‘minimum’: “A minimum score of 12 on the T-MMSE (which corresponds to a converted MMSE score of 16) was used as an inclusion criterion” (page 4, line 28).

4. The test battery is very sparse in some areas and has some odd choices for episodic memory (why no verbal memory?), working memory (digit span forward is not working memory, as there is no manipulation of information) and executive functioning (digit span backward is more accurately conceptualized as working memory, while semantic fluency is a language task).

We agree that the battery was not fully comprehensive and have deleted the sentence in our Methods that previously said, “In selecting tests, an important consideration was to sample cognitive domains representatively so that we could establish phenotypic profiles of deficits in the target neurodegenerative syndromes.” (page 6). We already direct readers to Table S1 which gives an overview of the tests that we did not choose to adopt for remote testing, along with a reason for why each test was not included.

We have changed ‘working memory’ to ‘Short-term verbal memory’ in Table 2 and Table S2, moved semantic fluency to the ‘Language’ domain in Table 2 and Table S2, and re-defined digit span backward as a ‘working memory’ task in Table 2 and Table S.

5. More detail is needed regarding the remote testing conditions. How many used a tablet vs laptop vs desktop computer? How many had someone else present in the room? What were the instructions for the remote test environment? Closed door? Pets? No sound? Turn off phones/devices? Did they have to show the examiner the environment to make sure there was no paper or calculator (for example)? Did distractions occur (e.g., interruptions from family, phone ringing?)

We appreciate this request and have added additional information about the remote testing conditions in Table 1 (page 5), and to Supplementary Material (page 1, lines 6-36; page) and Results (referencing at page 13, line 5, to the Supplementary Material page 2, lines 54-64). In short, a few distractions did occur, but these did not interfere significantly with administration of the remote neuropsychological battery.

6. A major threat to the generalizability of these results to clinical settings is that participants were recruited to a remote testing study rather than evaluating clinical participants who needed to be remote during covid to determine how these procedures would work in those who are not carefully selected/self-selected.

We agree that potential for generalizability of these results to clinical settings has not been proven, and this was beyond the scope of our study. To further protect against the possibility of misinterpretation, we have added a sentence to the paragraph on ‘limitations’ in the Discussion that says, “Finally, we note that the study was designed to explore the potential for remote neuropsychological assessments of research participants, and the results and conclusions here may not generalise to clinical settings.” (page 16, lines 48-50).

VERSION 2 – REVIEW

REVIEWER	Huygelier, Hanne KU Leuven
REVIEW RETURNED	20-Oct-2022

GENERAL COMMENTS	My concerns have been taken care of in the revised version of the manuscript.
---

REVIEWER	Chaytor, Naomi Washington State University
REVIEW RETURNED	17-Oct-2022

GENERAL COMMENTS	Thank you for addressing my initial review critique. I believe it is now ready for publication.
---